# Estimated Population Prevalence of Heart Failure with Reduced Ejection Fraction in Spain, According to DAPA-HF Study Criteria

**DOI:** 10.3390/jcm9072089

**Published:** 2020-07-03

**Authors:** Anna Camps-Vilaró, Juan F. Delgado-Jiménez, Núria Farré, Helena Tizón-Marcos, Jesús Álvarez-García, Juan Cinca, Irene R. Dégano, Jaume Marrugat

**Affiliations:** 1REGICOR Study Group, IMIM (Hospital del Mar Medical Research Institute), 08003 Barcelona, Spain; acamps@imim.es; 2CIBER of Cardiovascular Diseases (CIBERCV), Instituto de Salud Carlos III (ISCIII), 28029 Madrid, Spain; juan.delgado@salud.madrid.org (J.F.D.-J.); jalvarezg@santpau.cat (J.Á.-G.); JCinca@santpau.cat (J.C.); 3Department of Cardiology, Hospital Universitario 12 de Octubre, 28041 Madrid, Spain; 4Faculty of Medicine, Complutense University of Madrid (UCM), 28040 Madrid, Spain; 5Department of Cardiology, Hospital del Mar, 08003 Barcelona, Spain; NFarreLopez@parcdesalutmar.cat (N.F.); htizon@psmar.cat (H.T.-M.); 6Heart Diseases Biomedical Research Group (GREC), IMIM (Hospital del Mar Medical Research Institute), 08003 Barcelona, Spain; 7Faculty of Medicine, Universitat Autónoma de Barcelona (UAB), 08193 Barcelona, Spain; 8Department of Cardiology, Hospital de la Santa Creu i Sant Pau, 08041 Barcelona, Spain; 9Faculty of Medicine, University of Vic-Central University of Catalonia (UVic-UCC), 08500 Vic, Spain

**Keywords:** heart failure, reduced ejection fraction, cardiovascular disease, cardiovascular therapy, dapagliflozin, sodium-glucose cotransporter-2 inhibitor, epidemiology, prevalence

## Abstract

Heart failure (HF) is one of the main causes of morbidity, mortality, and high healthcare costs. Dapagliflozin, a sodium-glucose cotransporter-2 (SGLT2) inhibitor, reduced cardiovascular mortality and hospitalization for HF compared to placebo in patients with chronic HF, and reduced ejection fraction (EF) in the Dapagliflozin and Prevention of Adverse Outcomes in Heart Failure (DAPA-HF) study. Our aim was to estimate the number of patients with DAPA-HF characteristics in Spain. Our literature review identified epidemiological studies whose objective was to quantify the prevalence of HF and its comorbidities in Spain. We estimated the prevalence of HF with reduced EF, of New York Heart Association (NYHA) functional class II–IV, and with a glomerular filtration rate (GFR) ≥ 30 mL/min/1.73 m². In this population, we analysed the prevalence of diabetes using data from the REDINSCOR (Spanish Network for Heart Failure) registry. Our estimations indicate there are 594,684 patients ≥45 years old with HF in Spain (2.6% of this population age group), of which 52.4%, 84.0%, and 93.9% have reduced EF, are NYHA II–IV, and have a GFR ≥ 30 mL/min/1.73 m², respectively. By our calculations, approximately 245,789 Spanish patients would meet the DAPA-HF patient profile, and therefore could benefit from the protective cardiovascular effects of dapagliflozin.

## 1. Introduction

Heart failure (HF) is a clinical syndrome caused by a structural or functional heart abnormality leading to a reduction of cardiac output or an increase in intracardiac pressure. Characterization of HF is usually based on the ejection fraction (EF): preserved (≥50%), mid-range (40–49%), or reduced (<40%) [1].

Although HF is increasingly treatable and preventable, it remains one of the main causes of morbidity, mortality, and high healthcare costs. Incidence and prevalence of HF increase with population aging. Greater prevalence is related to increased survival, due to successful HF treatment, and higher incidence is related to the increasing population prevalence of atrial fibrillation, hypertension, obesity, and diabetes [1,2,3]. In 2018, HF caused 19,142 deaths in Spain [4]. While the morbidity and mortality of patients with reduced EF (40–60% of HF patients) [5,6,7] have decreased, thanks to improved clinical management in recent decades [8], the average total cost per patient with HF in Spain was €12,995–18,220 in 2010, with the largest cost item (59.1–69.8%) being non-professional care [9]. The total HF-patient healthcare expenditure as a percentage of the annual healthcare budget is 1.5% in all of Spain and 7.1% in Catalonia, with the highest cost in both cases related to hospitalizations [10,11].

Prevalence, clinical characteristics, and prognosis of HF differ according to EF category. Time series based on data from hospitalized patients suggest a more pronounced decrease in HF incidence and show higher all-cause mortality but fewer comorbidities in patients with reduced EF, compared to preserved EF [1].

Comorbidities are a key component in the management of patients with HF, as treatments for accompanying pathologies can worsen HF prognosis [1]. Two of the most common comorbidities are diabetes (20–40%) and chronic kidney disease (10–30%) [12,13]. Both are associated with more hospitalizations and higher mortality in HF patients [14].

The good quality of HF mortality data in Spain contrasts with the lesser-known prevalence of the disease. Several studies have analysed the population prevalence of HF in Spain, providing estimations that vary considerably (2.7–6.8%) for the population older than 44 years [14,15,16]. Most of these figures exceed the estimations in adult populations of Europe (1–2%) [1,17,18] and North America (2.2–2.4%) [2,19]. Moreover, no data are available on the prevalence of HF with reduced EF and comorbidities, the patient profile associated with the highest HF mortality.

Although advances in HF treatment have significantly reduced mortality risk in clinical trials, particularly in patients with reduced EF, the real-life impact of improved treatment has been more modest [20]. One new option is dapagliflozin, a sodium-glucose cotransporter-2 (SGLT2) inhibitor with solid clinical evidence in patients with type 2 diabetes mellitus (T2D) that also markedly reduces hospitalization for HF [21]. Dapagliflozin has reduced cardiovascular mortality and hospitalization for HF, compared to a placebo, as an additional treatment in patients with chronic HF and reduced EF, with and without T2D [22]. In view of these promising results, it would be useful to determine the target population for this new treatment indication, based on the criteria used in the Dapagliflozin and Prevention of Adverse Outcomes in Heart Failure (DAPA-HF) study [22]. Spain did not participate in the DAPA-HF study; therefore, there is a great interest to know the impact of the DAPA-HF study criteria in the Spanish population.

The aim of this study was to estimate the population in Spain, stratified by its autonomous communities, that meets the DAPA-HF criteria for dapagliflozin indication: HF and reduced EF (≤40%), functional class II–IV of the New York Heart Association (NYHA), and glomerular filtration rate (GFR) ≥ 30 mL/min/1.73 m², all with and without T2D.

## 2. Materials and Methods

### 2.1. Study Design and Population

To determine the number of patients with HF with reduced EF, in NYHA functional class II–IV, and with normal or moderate kidney function (≥ 30 mL/min/1.73 m²), we carried out a systematic literature search in PubMed of English or Spanish cross-sectional, cohort, population-based, or other epidemiological studies. To this end, we selected articles that contained the terms ‘‘heart failure’’ in the title; ‘‘prevalence’’ or ‘‘burden’’ in the title or abstract; ‘‘ejection fraction’’, ‘‘systole’’, or ‘‘systolic’’ in the title or abstract; and ‘‘general population’’ or ‘‘healthy’’ in the title or abstract (search date: September 1, 2017). This systematic search was updated on April 15, 2020, and supplemented with articles provided by experts [12,13,20,21,22,23,24].

Patients’ eligibility criteria for the population candidates for dapagliflozin treatment are summarized in Table 1. Full details are provided in the design paper [25].

### 2.2. Estimation of Population Prevalence and Number of Patients with Heart Failure in Spain

Our analysis applied the prevalence of HF reported for individuals older than 44 years in the largest and most recent representative population study [14] to the entire Spanish population and to each autonomous community’s population, according to the 2019 Spanish National Statistics Institute data [26].

### 2.3. Estimation of Ejection Fraction ≤ 40% and New York Heart Association Functional Class II–IV Prevalence in the Heart Failure Population in Spain

The proportion of HF patients with EF ≤ 40% and NYHA functional class II–IV was determined according to the average percentages observed in the most recent Spanish registries of patients with chronic HF [6,7,27,28,29]. The study designs, in reverse chronological order, are summarized as follows:

Prospective cohort studies:RICA (National Registry for Heart Failure), carried out in internal medicine units in 52 public and private hospitals [27];REDINSCOR (Spanish Network for Heart Failure), carried out in HF units in 18 hospitals [27];BADAPIC (Database of Patients with Heart Failure), involving 62 centres with specific HF units or clinics [6];

Cross-sectional studies:INCA (Heart Failure Study), involving 415 primary care physicians and 93 cardiologists [7];EPISERVE (Heart Failure in Outpatients), carried out by 181 primary care physicians, 172 cardiologists, and 154 internal medicine physicians [28];GALICAP (Galician Study of Heart Failure in Primary Care), involving 149 primary care physicians distributed in eight areas of Galicia [29].

### 2.4. Estimation of Glomerular Filtration Rate ≥ 30 mL/min/1.73 m² Prevalence in the Heart Failure Population in Spain

The proportion of patients with GFR ≥ 30 mL/min/1.73 m² was estimated from the most generalizable data, obtained from patients with chronic HF in 28 Spanish hospitals included in the European Society of Cardiology (ESC) Heart Failure Long-Term Registry [23]. The prevalence of GFR ≥ 30 mL/min/1.73 m² was applied to the estimated HF population with EF ≤ 40% and with NYHA functional class II–IV (obtained in Section 2.3).

A sensitivity analysis with the total REDINSCOR prevalence of EF, NYHA functional class II–IV, and GFR ≥ 30 mL/min/1.73 m² was done to determine the reliability of our estimates (Appendix A).

### 2.5. Estimation of Type 2 Diabetes Mellitus Prevalence in the Heart Failure Population in Spain

Finally, the prevalence of patients with and without T2D was obtained from a specific analysis of a chronic HF cohort provided by REDINSCOR investigators. The cohort characteristics have been described previously [27,30]. We calculated the prevalence of patients with and without diabetes who met the criteria for HF with EF ≤ 40%, NYHA functional class II–IV, and GFR ≥ 30 mL/min/1.73 m² (Appendix A).

### 2.6. Statistical Analysis

The arithmetic means of reduced EF and NYHA functional class II–IV estimations were obtained. The 95% confidence interval (CI) for the described estimations were obtained by assuming a Poisson distribution of patient counts. Analysis was carried out using R software, version 4.0.0. 

## 3. Results

### 3.1. Estimation of Population Prevalence and Number of Patients with Heart Failure in Spain

Table 2 describes the prevalence of HF in Spain and in Europe, as reported in the largest, most recent, and most representative population-based candidate studies. The 2012 study by Farré et al. [14] was considered the most appropriate for estimating prevalence in Spain, since it included a representative population of more than 88,000 people and the results closely resembled those of similar studies in Europe. Applying the prevalence of specific ages used by Farré et al. [14] to current Spanish population in the same age groups yielded a prevalence of approximately 2.6% in the group older than 44 years, or 594,684 (95% CI: 593,175–596,196) patients with HF.

### 3.2. Estimation of Ejection Fraction ≤ 40% and New York Heart Association Functional Class II–IV Prevalence in the Heart Failure Population in Spain

Characteristics of patients with HF included in the selected Spanish studies are shown in Table 3. The average proportion of HF patients with EF ≤ 40% was 52.4%, of which 84.0% had NYHA functional class II–IV. Applying these percentages to the estimated number of patients with HF in Spain, 311,614 (95% CI: 310,522–312,709) patients would have EF ≤ 40%, and 261,756 (95% CI: 260,755–262,760) patients would also have NYHA functional class II–IV.

The Spanish population and the number of patients with HF, by age group and by autonomous community, are shown in Table 4. In Table 5, these data are disaggregated by EF ≤ 40%, and by the combination of EF ≤ 40% and NYHA functional class II–IV.

### 3.3. Estimation of Glomerular Filtration Rate ≥ 30 mL/min/1.73 m² Prevalence in the Heart Failure Population in Spain

According to the prevalence of chronic HF patients with GFR ≥ 30 mL/min/1.73 m² described by Crespo-Lerio [23], we estimated the number of HF patients with the combination of EF ≤ 40%, NYHA functional class II–IV, and GFR ≥ 30 mL/min/1.73 m² (Table 6). In Spain, an estimated 245,789 (95% CI: 244,819–246,762) patients would have these characteristics. 

A sensitivity analysis with REDINSCOR estimates (Appendix A), however, showed that the estimated number of patients who met the clinical characteristics of participants in the DAPA-HF trial were 353,658 (95% CI: 352,494–354,825).

### 3.4. Estimation of Type 2 Diabetes Mellitus Prevalence in the Heart Failure Population in Spain

The number of HF patients with EF ≤ 40%, NYHA functional class II–IV, and GFR ≥ 30 mL/min/1.73 m², with and without T2D, are shown in Table 7 by age group and autonomous community. With these prevalence data, Spain would have 115,473 (95% CI: 114,809–116,140) patients with T2D and 130,316 (95% CI: 129,610–131,025) without T2D who would meet the diagnostic criteria of HF with EF ≤ 40%, NYHA functional class II–IV, and GFR ≥ 30 mL/min/1.73 m².

Figure 1 summarizes the sequence of patient selection by the prevalence of patient characteristics or comorbidities analysed, as well as the estimated number of patients with and without each characteristic in Spain.

## 4. Discussion

### 4.1. Main Findings

Among all Spanish patients with HF, the estimated prevalence of patients older than 44 years with reduced EF, NYHA functional class II–IV, and with normal or moderate kidney function (≥30 mL/min/1.73 m²) was about 41.3%. Since the population was selected to meet the clinical characteristics of participants in the DAPA-HF trial, it is conceivable that these patients might also benefit from the positive cardiovascular effects attributed to dapagliflozin, in addition to its glucose-lowering benefits. This new therapeutic indication would benefit the defined HF patient population, with and without T2D, as shown in the DAPA-HF clinical trial [22]. Regardless of the presence of T2D and the risk of worsening HF, death from cardiovascular causes and hospitalizations for HF were significantly less frequent among DAPA-HF participants who received dapagliflozin, compared to those who received a placebo. In the DAPA-HF study, dapagliflozin represents the first in a new class of drug for HF with reduced EF. The DAPA-HF study results introduce the opportunity to further study the potential cardiovascular benefits of SGLT2 inhibitors. Prior studies with empagliflozin [34] and canagliflozin [35] showed a reduction in the relative risk of HF hospitalization in T2D patients, suggesting that the observed benefit is not restricted to a particular drug, but is rather a class effect. The Canagliflozin Cardiovascular Assessment Study (CANVAS) showed that canagliflozin reduced the overall risk of HF events in patients with T2D and high cardiovascular risk. No clear difference in effects on HF with reduced versus preserved events was noted [36]. In addition, the Dapagliflozin Effect on Cardiovascular Events-Thrombolysis in Myocardial Infarction 58 (DECLARE-TIMI 58) study with dapagliflozin included a reanalysis of retrospectively-obtained EF. The clinical benefit of dapagliflozin was found to be strong in reduced EF in the subset of patient with available EF. In patients with HF without reduced EF, there was only a reduction of hospitalizations, but not in total or cardiovascular mortality [37]. The former was confirmed in the DAPA-HF study. The dapagliflozin effect on mortality in HF-preserved EF patients remains to be conclusively answered

### 4.2. Reliability of Prevalence Estimates

Based on updated demographic information and a comprehensive literature review to obtain reliable prevalence data, we selected the most recent population-based cross-sectional study, in which Farré et al. analysed data from 88,000 individuals representative of the population of Catalonia, with an estimated a 2.7% HF prevalence in people older than 44 years [14]. Due to slight regional differences in age group population distribution, our estimations yielded 2.6% prevalence when applied to all of Spain. The estimations obtained by Farré et al. are also close to European and North American figures, and are more recent than the larger prevalence estimates reported in the meta-analysis by Hernáez et al. [24] and the 2008 Heart Failure Prevalence Study in Spain (PRICE) by Anguita et al. [15].

Given the diversity of hospital departments and primary care settings participating in the published studies, our estimated prevalence of HF patients with reduced EF and NYHA functional class II–IV (52.4% and 84.0%, respectively) reflects the mean values of the main published Spanish registries [6,7,27,28,29]. The obtained prevalence values are in good agreement with the most recent European publications [5,38] in the Southern Europe population. Spanish cardiology departments were well-represented in these European studies, carried out in the context of the ESC Heart Failure Long-Term Registry, which had indicated that the prevalence of reduced EF in patients with chronic HF predominantly admitted to cardiology departments would be around 60%. This is slightly higher than our study’s estimate, which was based on HF data from cardiology units, internal medicine, and primary care settings. The Linx Registry, one of the most recent HF studies in Spain, also had a large sample of patients and was carried out in cardiology departments in Catalonia. In that registry, de Frutos et al. [39] estimated that the prevalence of NYHA functional class II–IV in patients with HF and EF ≤ 40% was 85.5%, remarkably close to the estimate obtained in the present study.

We used the 6.1% prevalence of GFR < 30 mL/min/1.73 m² among chronic HF patients, reported by Crespo-Leiro et al. [23], based on data from 28 Spanish hospitals included in the ESC Heart Failure Long-Term Registry. This is the most recent publication with HF data from Spain, includes the largest series of Spanish patients, and is the only study to provide GFR data generalizable to the HF population. The 6.1% prevalence is among the lowest published in recent decades in Spain, although it is close to the REDINSCOR registry prevalence of 5.5% in HF units within cardiology departments (Appendix A).

### 4.3. Potential Translational Perspective

The results of the DAPA-HF study demonstrated that the primary composite outcome occurred in 386 of 2373 patients (16.3%) in the dapagliflozin group and in 502 of 2371 patients (21.2%) in the placebo group (hazard ratio = 0.74; 95% CI: 0.65–0.85; *p* < 0.001). The largest number of events of worsening HF was hospitalizations. Of the patients receiving dapagliflozin, 231 (9.7%) were hospitalized for HF, compared with 318 patients (13.4%) receiving the placebo (hazard ratio = 0.70; 95% CI: 0.59–0.83). Death from cardiovascular causes occurred in 227 patients (9.6%) who received dapagliflozin, and in 273 (11.5%) who received the placebo (hazard ratio = 0.82; 95% CI: 0.69 to 0.98) [22]. Under the placebo group incidence assumptions, among all 245,789 patients of the estimated Spanish target population, the primary outcome would occur in 52,107 patients, 32,936 patients would be hospitalized for HF, and 28,266 patients would die from cardiovascular causes. With dapagliflozin therapy, the expected annual reduction would consist of 5996 hospitalized patients for HF and 3079 deaths from cardiovascular causes.

### 4.4. Strengths and Limitations

The main strength of this study is that it combines the estimates from recently published HF prevalence data, enriched by a specific analysis of the REDINSCOR registry database to estimate HF with reduced EF, NYHA functional class II–IV, and GFR ≥ 30 mL/min/1.73 m², with and without T2D prevalence in the Spanish population. However, we decided not to use the estimates based on REDINSCOR database, because the REDINSCOR registry could be biased toward the profile of patients admitted to HF units in cardiology departments, which could depart from the general HF patient population characteristics. All patients included in the REDINSCOR registry had a NYHA class >I, and the mean estimate of patients with reduced EF was 73%, considerably different from the average of the main Spanish registries that we summarize in Table 3. A sensitivity analysis with REDINSCOR patient characteristics is presented in Appendix A, showing rather higher figures than its corresponding Table 6 results.

The study also has several limitations. First, we did not take into account the N-terminal pro B-type natriuretic peptide (NT-proBNP) eligibility criteria prevalence in our estimates, due to the absence of prevalence information in the literature and to the probably small reduction in the number of eligibility patients. Second, among the Spanish studies summarized in Table 3, the definition of reduced EF varied from EF ≤ 40% to EF ≤ 50%. Furthermore, the GALICAP and EPISERVE studies did not differentiate between chronic and acute HF. Third, we had to assume that the prevalence of NYHA functional class II–IV was the same for reduced, mid-range, and preserved EF, owing to the absence of stratified information in the literature. Likewise, we had to assume that the prevalence of GFR ≥ 30 mL/min/1.73 m² was the same for each type of EF and NYHA functional class, and for all age groups considered in our study. Fourth, the prevalence of diabetic and non-diabetic patients in the REDINSCOR registry could be biased toward the profile of patients admitted to HF units in cardiology departments. Finally, our study does not report prevalence by sex, as the published studies did not provide this stratified information. We firmly support stratification by sex in all future studies, in order to identify the best treatment guidelines to apply in the whole population.

## 5. Conclusions

In this population analysis, we estimated that approximately 245,789 Spanish patients would meet the inclusion criteria of the DAPA-HF: EF ≤ 40%, NYHA functional class II–IV, and GFR ≥ 30 mL/min/1.73 m², as well as 115,473 with T2D. The magnitude of this population highlights the need to introduce effective and safe new drugs to reduce morbidity and mortality in these patients.

## Figures and Tables

**Figure 1 jcm-09-02089-f001:**
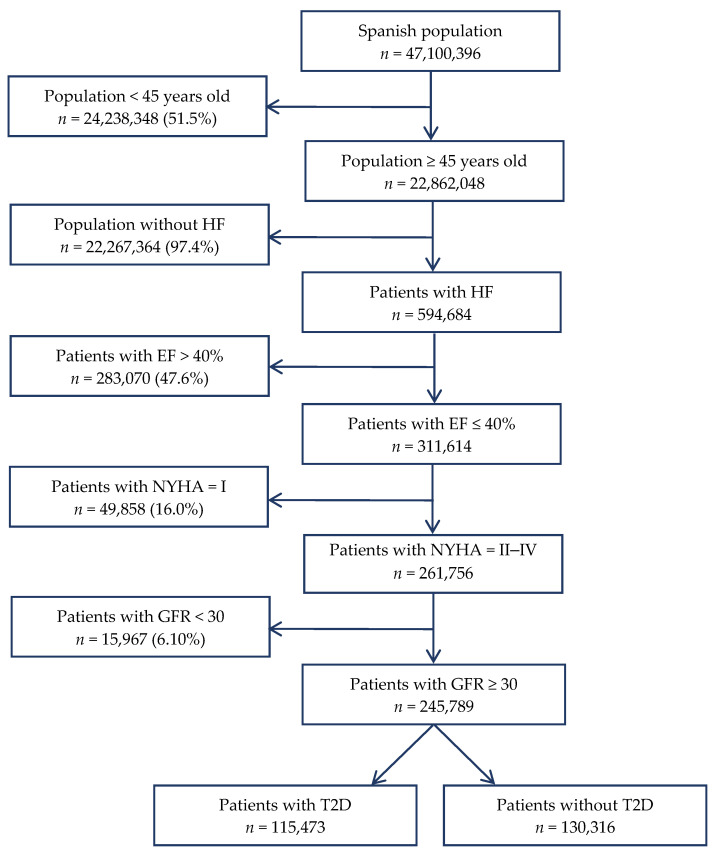
Flow chart of the Spanish population older than 44 years with heart failure (HF), ejection fraction (EF) ≤ 40%, New York Heart Association (NYHA) functional class II–IV, glomerular filtration rate (GFR) ≥ 30 mL/min/1.73 m², and type 2 diabetes mellitus (T2D), according to the estimation specified in the Methods section.

**Table 1 jcm-09-02089-t001:** Summary of inclusion and exclusion criteria of the DAPA-HF study [25].

**Inclusion Criteria**
(1) Provision of signed informed consent prior to any study specific procedures
(2) Men or women, aged ≥18 years at the time of consent
(3) Diagnosis of HF with left ventricular EF ≤ 40%, which has been present for at least 12 months prior to enrolment
(4) Diagnosis of symptomatic HF (NYHA functional class II–IV), within the previous 2 months
(5) Optimally treated with pharmacological and/or device therapy for HF
(6) NT-proBNP ≥ 600 pg/mL (or if hospitalised for HF within the previous 12 months, NT-proBNP ≥ 400 pg/mL) at enrolment. Patients with atrial fibrillation or atrial flutter must have a level ≥900 pg/mL, irrespective of history of HF hospitalization
**Exclusion Criteria**
(1) Treatment with SGLT2 inhibitors within 8 weeks prior to enrolment, or previous intolerance of an SGLT2 inhibitor
(2) Diagnosis of type 1 diabetes mellitus
(3) Symptomatic hypotension or systolic blood pressure <95 mmHg
(4) Recent worsening HF or other cardiovascular events or procedures
(5) Severe (GFR < 30 mL/min/1.73m² by CKD-EPI equation), unstable, or rapidly progressing renal disease at the time of randomization
(6) Other conditions likely to prevent patient participation in the trial or greatly limit life expectancy

DAPA-HF: Dapagliflozin and Prevention of Adverse Outcomes in Heart Failure; HF: heart failure; EF: ejection fraction; NYHA: New York Heart Association; NT-proBNP: N-terminal proB-type natriuretic peptide; GFR: glomerular filtration rate; CKD-EPI: Chronic Kidney Disease-Epidemiology Collaboration; SGLT2: sodium-glucose cotransporter-2.

**Table 2 jcm-09-02089-t002:** Summary of the heart failure prevalence by age group in population studies in Spain and Europe.

	Year of Publication	Country/Region	Study Population	Age, Years (SD)	Women,%	Age Group Prevalence, By Years						
						Total	45–54	55–64	65–74	≥75		
Farré et al. [14]	2017	Catalonia	88,195	77 (12)	55.0%	2.70%	0.30%	0.90%	2.50%	8.80%		
Anguita et al. [15]	2008	Spain	1776	64 (12)	55.9%	6.80%	1.30%	5.50%	8.00%	16.1%		
							40–49	50–59	60–69	70–79	≥80	
Cortina et al. [16]	2001	Asturias	391	60 (13)	53.6%	5.00%	<1.00%	2.00%	5.00%	13.0%	18.0%	
							0–49	50–59	60–69	70–79	80–89	≥90
Ohlmeier et al. [17]	2015	Germany	6,284,194	39 (21)	48.2%	1.70%	0.10%	1.28%	3.88%	10.8%	25.9%	45.7%
							19–54	55–64	65–74	75–84	85–99	
Parén et al. [18]	2014	Sweden	144,925	-	48.9%	1.99%	0.13%	1.06%	3.20%	9.23%	20.4%	
							45–64		65–74	≥75		
Murphy et al. [31]	2004	Scotland	307,741	-	50.6%	0.71%	0.38%		2.30%	6.53%		
								55–64	65–74	75–84	≥85	
Bleumink et al. [32]	2004	Rotterdam	7983	70 (10)	61.0%	6.70%	-	0.90%	4.00%	9.70%	17.4%	
							25–49	50–59	60–69	70–79	≥80	
Ceia et al. [33]	2002	Portugal	5434	68 (15)	62.7%	4.36%	1.36%	2.93%	7.63%	12.7%	16.1%	

SD: standard deviation.

**Table 3 jcm-09-02089-t003:** Studies of patients with heart failure in Spain.

Study	Year of Publication	Study Population	Age, Years (SD)	Women, %	Type of Heart Failure	Geographical Area	Specialty	Reduced EF	NYHA Class II–IV
RICA [27]	2017	1396	79 (8)	54.0%	chronic	Spain	Internal medicine	39.5%	80.2%
REDINSCOR [27]	2017	2150	70 (10)	31.0%	chronic	Spain	Cardiology	79.6%	100% ¹
BADAPIC [6]	2010	4720	65 (12)	29.0%	chronic	Spain	Cardiology	63.4%	100% ¹
INCA [7]	2009	2161	71 (11)	44.4%	chronic	Spain	Primary care and cardiology	38.3%	84.0%
EPISERVE [28]	2008	2249	72 (10)	45.0%	-	Spain, except La Rioja	Primary care, cardiology, and internal medicine	55.0%	90.4%
GALICAP [29]	2007	1195	76 (10)	52.1%	-	Galicia	Primary care	38.6%	81.5%
Mean								52.4%	84.0%

¹ Data not included in the calculation of NYHA functional class II–IV average. RICA: National Registry for Heart Failure; REDINSCOR: Spanish Network for Heart Failure; BADAPIC: Database of Patients with Heart Failure; INCA: Heart Failure Study; EPISERVE: Heart Failure in Outpatients; GALICAP: Galician Study of Heart Failure in Primary Care; SD: standard deviation; Reduced EF: ejection fraction less than 40–50%; NYHA: New York Heart Association.

**Table 4 jcm-09-02089-t004:** Total population in Spain according to 2019 data from the National Statistics Institute [26], and the number of patients with heart failure in Spain and its autonomous communities, according to prevalence data published by Farré et al. [14].

	General Population (by Age Group)					Heart Failure Prevalence (by Age Group)				
	Total	45–54	55–64	65–74		Total	45–54	55–64	65–74	75+
Autonomous Communities							0.30%	0.90%	2.50%	8.80%
Andalucía	3,895,025	1,341,487	1,093,149	770,187	690,202	93,855	4024	9838	19,255	60,738
Aragón	674,882	207,355	180,224	136,185	151,118	18,947	622	1622	3405	13,298
Principado de Asturias	589,263	164,027	161,384	131,949	131,903	16,851	492	1452	3299	11,607
Illes Balears	529,589	194,110	145,007	103,545	86,927	12,126	582	1305	2589	7650
Canarias	1,048,012	390,823	293,621	199,438	164,130	23,244	1172	2643	4986	14,443
Cantabria	307,255	93,358	85,403	65,351	63,143	8239	280	769	1634	5557
Castilla y León	1,347,618	376,632	361,698	275,863	333,425	40,623	1130	3255	6897	29,341
Castilla-La Mancha	974,147	320,338	265,833	180,457	207,519	26,127	961	2392	4511	18,262
Catalunya	3,593,591	1,188,206	946,935	738,057	720,393	93,933	3565	8522	18,451	63,395
Comunitat Valenciana	2,437,743	799,940	658,274	508,440	471,089	62,491	2400	5924	12,711	41,456
Extremadura	537,686	166,131	151,011	104,505	116,039	14,682	498	1359	2613	10,211
Galicia	1,484,698	424,418	377,370	322,257	360,653	44,463	1273	3396	8056	31,737
Comunidad de Madrid	3,099,482	1,074,013	826,152	609,975	589,342	77,769	3222	7435	15,249	51,862
Región de Murcia	648,115	234,820	176,760	123,046	113,489	15,358	704	1591	3076	9987
Comunidad Foral de Navarra	317,203	102,519	84,911	64,649	65,124	8419	308	764	1616	5731
País Vasco	1,156,253	347,445	312,977	247,706	248,125	31,887	1042	2817	6193	21,835
La Rioja	158,846	49,143	43,080	32,058	34,565	4378	147	388	801	3042
Ciudad Autónoma de Ceuta	32,632	12,059	10,246	5676	4651	680	36	92	142	409
Ciudad Autónoma de Melilla	30,008	11,057	9737	5077	4137	612	33	88	127	364
All Communities	22,862,048	7,497,881	6,183,772	4,624,421	4,555,974	594,684	22,494	55,654	115,611	400,926
95% CI	22,852,679–22,871,420	7,492,516–7,503,249	6,178,900–6,188,647	4,620,208–4,628,637	4,551,792–4,560,158	593,175–596,196	22,202–22,789	55,194–56,117	114,947–116,278	399,687–402,168

CI: Confidence interval.

**Table 5 jcm-09-02089-t005:** Total number of patients in Spain and its autonomous communities with heart failure and ejection fraction ≤ 40%, as well as the number of heart failure patients with average ejection fraction ≤ 40% and in New York Heart Association functional class II–IV (based on data summarized in Table 3).

	Heart Failure + Ejection Fraction ≤ 40% (by Age Group)					Heart Failure + Ejection Fraction ≤ 40% + NYHA Class II–IV (by Age Group)				
	Total	45–54	55–64	65–74	75+	Total	45–54	55–64	65–74	75+
Autonomous Communities		52.4%	52.4%	52.4%	52.4%		84.0%	84.0%	84.0%	84.0%
Andalucía	49,180	2109	5155	10,089	31,827	41,311	1771	4330	8475	26,734
Aragón	9928	326	850	1784	6968	8340	274	714	1499	5853
Principado de Asturias	8830	258	761	1729	6082	7417	217	639	1452	5109
Illes Balears	6354	305	684	1356	4008	5337	256	574	1139	3367
Canarias	12,180	614	1385	2613	7568	10,231	516	1163	2195	6357
Cantabria	4317	147	403	856	2912	3627	123	338	719	2446
Castilla y León	21,287	592	1706	3614	15,375	17,881	497	1433	3036	12,915
Castilla-La Mancha	13,690	504	1254	2364	9569	11,500	423	1053	1986	8038
Catalunya	49,221	1868	4466	9669	33,219	41,346	1569	3751	8122	27,904
Comunitat Valenciana	32,745	1258	3104	6661	21,723	27,506	1056	2608	5595	18,247
Extremadura	7693	261	712	1369	5351	6462	219	598	1150	4495
Galicia	23,299	667	1780	4222	16,630	19,571	560	1495	3546	13,970
Comunidad de Madrid	40,751	1688	3896	7991	27,176	34,231	1418	3273	6712	22,828
Región de Murcia	8048	369	834	1612	5233	6760	310	700	1354	4396
Comunidad Foral de Navarra	4411	161	400	847	3003	3706	135	336	711	2523
País Vasco	16,709	546	1476	3245	11,442	14,035	459	1240	2726	9611
La Rioja	2294	77	203	420	1594	1927	65	171	353	1339
Ciudad Autónoma de Ceuta	356	19	48	74	214	299	16	41	62	180
Ciudad Autónoma de Melilla	321	17	46	67	191	269	15	39	56	160
All Communities	311,614	11,787	29,163	60,580	210,085	261,756	9901	24,497	50,887	176,471
95% CI	310,522–312,709	11,576–12,001	28,830–29,499	60,100– 61,063	209,189–210,984	260,755–262,760	9708–10,097	24,192–24,805	50,447–51,330	175,650–177,295

NYHA: New York Heart Association; CI: Confidence interval.

**Table 6 jcm-09-02089-t006:** Total number of patients with heart failure ejection fraction ≤ 40%, New York Heart Association functional class II–IV, and glomerular filtration rate ≥ 30 mL/min/1.73 m² in Spain and its autonomous communities [23].

	HF + EF ≤ 40% + NYHA Class II–IV + GFR ≥ 30 (by Age Group)				
	Total	45–54	55–64	65–74	75+
Autonomous Communities		93.9%	93.9%	93.9%	93.9%
Andalucía	38,791	1663	4066	7958	25,104
Aragón	7831	257	670	1407	5496
Principado de Asturias	6965	203	600	1363	4797
Illes Balears	5012	241	539	1070	3162
Canarias	9607	485	1092	2061	5970
Cantabria	3405	116	318	675	2297
Castilla y León	16,790	467	1345	2850	12,127
Castilla-La Mancha	10,798	397	989	1865	7548
Catalunya	38,823	1473	3522	7626	26,202
Comunitat Valenciana	25,828	992	2449	5254	17,134
Extremadura	6068	206	562	1080	4220
Galicia	18,377	526	1404	3330	13,117
Comunidad de Madrid	32,143	1332	3073	6303	21,435
Región de Murcia	6348	291	658	1271	4128
Comunidad Foral de Navarra	3480	127	316	668	2369
País Vasco	13,179	431	1164	2559	9025
La Rioja	1810	61	160	331	1257
Ciudad Autónoma de Ceuta	281	15	38	59	169
Ciudad Autónoma de Melilla	253	14	36	52	150
All Communities	245,789	9297	23,002	47,783	165,707
95% CI	244,819–246,762	9,110–9,487	22,707–23,300	47,357–48,212	164,911–166,506

HF: heart failure; EF: ejection fraction; NYHA: New York Heart Association; GFR: glomerular filtration rate. (mL/min/1.73 m²); CI: Confidence interval.

**Table 7 jcm-09-02089-t007:** Total number of patients with heart failure, ejection fraction ≤ 40%, New York Heart Association functional class II–IV, and glomerular filtration rate ≥ 30 mL/min/1.73 m², with or without type 2 diabetes mellitus, in Spain and its autonomous communities, according to prevalence data from the REDINSCOR registry (Appendix A).

	HF + EF ≤ 40% + NYHA Class II–IV + GFR ≥ 30 + Diabetes (by Age Group)					HF + EF ≤ 40% + NYHA Class II–IV + GFR ≥ 30 (No Diabetes) (by Age Group)				
	Total	45–54	55–64	65–74	75+	Total	45–54	55–64	65–74	75+
Autonomous Communities		28.7%	40.7%	49.2%	48.2%		71.3%	59.3%	50.8%	51.8%
Andalucía	18,157	478	1654	3914	12,111	20,634	1185	2412	4044	12,993
Aragón	3690	74	273	692	2652	4141	183	398	715	2845
Principado de Asturias	3288	58	244	671	2314	3677	145	356	693	2483
Illes Balears	2340	69	219	526	1525	2672	172	320	544	1636
Canarias	4477	139	444	1014	2880	5130	345	648	1047	3090
Cantabria	1603	33	129	332	1108	1803	82	188	343	1189
Castilla y León	7934	134	547	1402	5851	8856	333	798	1448	6277
Castilla-La Mancha	5075	114	402	917	3641	5724	283	587	947	3906
Catalunya	18,248	423	1433	3751	12,641	20,576	1050	2090	3875	13,561
Comunitat Valenciana	12,131	285	996	2584	8266	13,697	707	1453	2670	8868
Extremadura	2855	59	228	531	2036	3213	147	333	549	2184
Galicia	8688	151	571	1638	6328	9689	375	833	1692	6789
Comunidad de Madrid	15,074	383	1250	3100	10,341	17,069	949	1823	3203	11,094
Región de Murcia	2968	84	267	625	1991	3380	207	390	646	2136
Comunidad Foral de Navarra	1636	37	128	329	1143	1843	91	187	339	1226
País Vasco	6210	124	473	1259	4,354	6969	307	691	1301	4671
La Rioja	852	18	65	163	607	957	43	95	168	651
Ciudad Autónoma de Ceuta	130	4	16	29	82	151	11	23	30	88
Ciudad Autónoma de Melilla	117	4	15	26	73	136	10	21	27	78
All Communities	115,473	2672	9355	23,503	79,943	130,316	6625	13,647	24,280	85,764
95% CI	114,809–116,140	2573–2774	9167–9546	23,205–23,804	79,391–80,498	129,610–131,025	6467–6786	13,420–13,877	23,977–24,586	85,192–86,339

HF: heart failure; EF: ejection fraction; NYHA: New York Heart Association; GFR: glomerular filtration rate (mL/min/1.73 m²); CI: Confidence interval.

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
