# Peer review of "Estimated Population Prevalence of Heart Failure with Reduced Ejection Fraction in Spain, According to DAPA-HF Study Criteria"

_jcm, 2020, doi:10.3390/jcm9072089_

Round 1
Reviewer 1 Report
The manuscript submitted by Vilaro et al. describes the criteria used to find out prevalent heart failure patients in a given population. Specifically, the authors describe the patients with heart failure, ejection fraction ≤40%, New York Heart Association functional class II–IV, glomerular filtration rate ≥30 mL/min/1.73 m2, with or without type 2 diabetes mellitus needs Dapagliflozin as a promising treatment. This population- to individual-based approach and phenotypic characterization of each patient or potential patient will help in prognosis of heart failure patients. I have some concerns that additional figures are needed to strengthen the study and think this should be considered/acknowledged. Nevertheless, this study is interesting, logical, and well-supported by its data, and would be of interest to the clinical cardiology fields.
- Authors needs additional table or figure to describe the DAPA-HF patients’ selection (key inclusion) criteria used in Spain.
- Authors need to explain the current status of DAPA-HF trial in Spain.
- What other possible drugs can be used for this patient population other than Dapagliflozin.
- Please provide speculation for the treatment of heart failure with preserved ejection fraction and type2 diabetes patients.
- A translational perspective paragraph is important for the study.
Reviewer 2 Report
The authors present an analysis of the crude estimated number of patients in Spain who meet the entry criteria of the DAPA-HF trial. The analysis leverages estimated from national databases to estimate how many people may be eligible for treatment with dapagliflozin in HFrEF. I have the following comments:
- Could the authors please describe the literature review methods used.
- The authors have made the assumption that distribution of ejection fraction, NYHA, eGFR are the same across age-groups. For example – HFrEF is the predominant phenotype in younger patients with the proportion of patients with HFpEF increasing with increasing. Moreover, eGFR is a function of age. Can the authors provide a more granular analysis taking into account the age variation in these variables – they appear to have these data available at least in the REDINSCOR database (table S1 and S2). A sensitivity analysis of the estimates based on this database would be of interest.
- It seems a shame to me to put together this work and then not estimate the potential treatment benefits in the population at a national level if dapagliflozin was used – have the authors considered this?
- The authors don’t comment on the fact they have not taken into account the NTproBNP inclusion criteria in DAPA-HF (or any of the exclusion criteria such as recent MI, recent hospitalisation for HF etc.) This should at least be mentioned.
Round 2
Reviewer 2 Report
The authors have improved the quality of the manuscript and have adequately addressed my comments.